METHODS AND RESOURCES

# Cartography of teneurin and latrophilin expression reveals spatiotemporal axis heterogeneity in the mouse hippocampus during development

Kif Liakath-Ali [1]¤*, Rebecca Refaee[1], Thomas C. Südhof[1,2]*

1 Department of Molecular and Cellular Physiology, Stanford University, Stanford, California, United States of America, 2 Howard Hughes Medical Institute, Stanford University, Stanford, California, United States of America

¤ Current address: School of Biological Sciences, University of Southampton, Southampton, United Kingdom
* kif@stanford.edu, kif.liakath-ali@soton.ac.uk (KL-A); tcs1@stanford.edu (TCS)

**Data Availability Statement:** All relevant data are within the paper and its Supporting Information files.

## Abstract

Synaptic adhesion molecules (SAMs) are evolutionarily conserved proteins that play an important role in the form and function of neuronal synapses. Teneurins (Tenms) and latrophilins (Lphns) are well-known cell adhesion molecules that form a transsynaptic complex. Recent studies suggest that *Tenm3* and Lphn2 (gene symbol *Adgrl2*) are involved in hippocampal circuit assembly via their topographical expression. However, it is not known whether other teneurins and latrophilins display similar topographically restricted expression patterns during embryonic and postnatal development. Here, we reveal the cartography of all teneurin (*Tenm1-4*) and latrophilin (Lphn1-3 [*Adgrl1-3*]) paralog expression in the mouse hippocampus across prenatal and postnatal development as monitored by large-scale single-molecule RNA in situ hybridization mapping. Our results identify a striking heterogeneity in teneurin and latrophilin expression along the spatiotemporal axis of the hippocampus. Tenm2 and Tenm4 expression levels peak at the neonatal stage when compared to Tenm1 and Tenm3, while Tenm1 expression is restricted to the postnatal pyramidal cell layer. Tenm4 expression in the dentate gyrus (DG) exhibits an opposing topographical expression pattern in the embryonic and neonatal hippocampus. Our findings were validated by analyses of multiple RNA-seq datasets at bulk, single-cell, and spatial levels. Thus, our study presents a comprehensive spatiotemporal map of Tenm and Lphn expression in the hippocampus, showcasing their diverse expression patterns across developmental stages in distinct spatial axes.

## Introduction

Synaptic adhesion molecules (SAMs) are a diverse family of transmembrane proteins that play crucial roles in the development, maintenance, and plasticity of synapses in the nervous system [1]. Interactions of SAMs contribute to the stabilization of synapses, regulation of

**Funding:** Financial support from the National Institute of Mental Health (MH052804 and MH116529) (https://www.nimh.nih.gov/) to TCS and the European Molecular Biology Organization (https://www.embo.org/) Long Term Fellowship (ALTF 803-2017) and the Larry L. Hillblom Foundation (https://llhf.org/) Fellowship grant (2020-A-016-FEL) to KL-A are gratefully acknowledged. The funders had no role in study design, data collection and analysis, decision to publish, or preparation of the manuscript.

**Competing interests:** The authors have declared that no competing interests exist.

**Abbreviations:** ADHD, attention deficit hyperactivity disorder; ASCOT, alternative splicing catalog of the transcriptome; CNS, central nervous system; DG, dentate gyrus; EC, entorhinal cortex; GCL, granule cell layer; GPCR, G protein-coupled receptor; PBS, phosphate-buffered saline; PT, pyramidal track; ROI, region of interest; SAM, synaptic adhesion molecule; scRNA-seq, single cell RNA-seq; smRNA-FISH, single-molecule RNA fluorescent in situ hybridization; SVZ, subventricular zone.

neurotransmitter release, and modulation of synaptic strength. Dysfunction of SAMs has been implicated in many neurological and psychiatric disorders, highlighting their crucial role in nervous system function. Among well-known SAMs, teneurins (Tenm or Odz) and latrophilins (Lphn or Adgrl) play an important role in the evolutionary organization of the central nervous system (CNS) across diverse species [2]. Tenm and Lphn interact as ligand and receptor to form a transsynaptic complex, which is essential for brain development and the maintenance of functional neural circuits in the brain [3–5].

Teneurins are single-pass transmembrane proteins that are expressed across multiple tissues in mammals with the highest levels detected in the brain. Mammals generally express four Tenm paralogs (*Tenm1* to *Tenm4*), whereas arthropods have 2 Tenm paralogs, and other invertebrates have only 1. Teneurins are composed of a large extracellular sequence, a single transmembrane region, and an N-terminal cytoplasmic sequence. In brain development, Tenms are thought to regulate neuronal migration, act as axon guidance molecules, and specify synapse formation [6–9]. Deletion of Tenms is known to cause synapse loss and perturb synaptic organization [6,10]. Similar to other SAMs, teneurins likely act as signaling molecules in their various functions by activating intracellular signaling pathways that in turn affect synaptic transmission and plasticity [11]. For example, teneurins have been suggested to modulate presynaptic calcium influx and neurotransmitter release, as well as postsynaptic receptor trafficking and localization [5]. These unique properties of teneurins make them important players in the intricate process of neuronal communication [12,13].

Lphns, on the other hand, are seven transmembrane domain-containing G protein-coupled receptors (GPCRs) with a long N-terminal extracellular sequence that are thought to play a similar role as Tenms in organismal and neuronal development [14]. The three Lphn paralogs in mammals (*Lphn1* to *Lphn3*) are widely expressed in the cells of the nervous, immune, and digestive systems. Lphns were originally identified as calcium-independent receptors for α-latrotoxin, a black widow spider toxin that triggers massive neurotransmitter release from nerve terminals [2]. The developmental role of Lphns has been studied extensively in worms and drosophila. In mice, the extracellular regions of Lphns mediate excitatory synapse formation [15]. Among the many candidate SAMs involved in synapse formation, deletion of Lphn2 causes one of the strongest synapse-formation phenotypes [16]. Lphn mutations have been linked to attention deficit hyperactivity disorder (ADHD) and various cancers in humans [17]. Moreover, it was recently shown that complementary expression pattern of Tenm3 and Lphn2 in the mediolateral axis of the hippocampus enables the assembly of CA1 to subiculum network [18]. Both Tenms and Lphns are expressed in alternatively spliced isoforms, adding additional layers of complexity to the regulation of their various canonical and noncanonical functions [2,3,19].

The development of the mouse hippocampus begins during embryogenesis and extends through postnatal stages. While neurogenesis is indeed a prominent feature of the adult hippocampus, particularly in the dentate gyrus (DG) and subventricular zone (SVZ), it is crucial to distinguish this from embryonic hippocampal development. During embryonic stages, the primordial hippocampal area, known as the hippocampal primordium, arises from the ventral telencephalon [20]. Origin of hippocampal neurons occurs primarily through local proliferation of progenitors within the ventricular zone of this region. These progenitors give rise to glutamatergic neurons destined for various hippocampal layers, including the granule cell layer (GCL) of the DG. Interneurons, on the other hand, originate from separate embryonic structures like the medial ganglionic eminence and migrate into the developing hippocampus [21]. It is important to note that the DG itself emerges slightly later in development, with its characteristic GCL forming postnatally in most rodents. In rats, for example, the vast majority of DG granule cells are born after birth [22]. Maturation of the hippocampus continues

throughout early postnatal life, with the establishment of neural circuits and synaptic connections critical for proper functioning [23,24].

The division of dorsal and ventral hippocampus is based on anatomical and functional differences. This idea of dichotomy of hippocampal function based on earlier studies that showed distinct input and output connections in the ventral and dorsal hippocampus, dependence of spatial memory on the dorsal hippocampus and emotional behavior and affective processes on the ventral hippocampus [25–28]. The role of the ventral hippocampus has been implicated in anxiety-related behaviors, stress responses, and regulation of the hypothalamic–pituitary–adrenal axis. It is connected to brain regions associated with emotional processing, such as the amygdala and prefrontal cortex. Dysfunction in the ventral hippocampus has been linked to psychiatric disorders, including anxiety and mood disorders. Conversely, the dorsal hippocampus is primarily associated with cognitive functions, including spatial learning and memory, and plays a critical role in spatial navigation and the formation of cognitive maps. It is extensively connected with brain regions involved in memory processing, such as the entorhinal cortex, prefrontal cortex, and subiculum [25,29,30].

Recent studies have shown that *Tenm3* and *Lphn2* are expressed in multiple topographically interconnected regions of hippocampal circuits [18,31,32]. Co-expression patterns of Tenms and Lphns during organ development and in the brain suggest that their respective functions in embryonic and neural development may be mediated via their heterophilic interaction. However, an extensive expression map of all Tenm and Lphn paralogs is not available at the spatiotemporal level within the hippocampal circuit (S1A Fig). This "cartography" is essential to understand the spatial distribution linking the functional roles of Tenm and Lphn paralogs. In this present study, we aimed to map the mRNA expression of Tenm (1–4) and Lphn (1–3) paralogs and validated it through multiple public RNA-seq datasets. Our results show striking spatial and developmental dynamics of Tenm and Lphn expression in the hippocampal circuit and suggest that their spatial and cell type-specific distribution might underlie their functional significance.

## Results

In rodents, it is known that the extrinsic connectivity of the hippocampus is organized in smooth topographical gradients [30]. The longitudinal axis of the hippocampus is well-captured in sagittal section planes (S1B Fig) that reveal the cells of all pathways (Schaffer collaterals, perforant, mossy fibers, and temporoammonic) in the hippocampal circuit. Therefore, we chose to map and quantify Tenm and Lphn mRNA expression patterns in 4 representative sagittal (lateromedial) cross sections of the mouse brain at embryonic day 16.5 (E16.5) and postnatal days 0, 10, and 30 (P0, P10, P30) (Fig 1). In addition to this, we also mapped the Tenm and Lphn expression patterns in horizontal section planes but did not quantify the expression patterns in these planes because no reference atlas for horizontal sections is available.

Cumulative intensities across all lateromedial sections showed a striking absence of Tenm1 in E16.5 and P0 hippocampal regions. Tenm2 and Tenm4 showed higher expression levels when compared to Tenm3 (Figs 2A–2D and S2–S5). At P10 and P30, Tenm1 is expressed at high levels in the stratum pyramidale (s.p.) layer of CA1 compared to CA3 regions. In addition, Tenm1 is synthesized in all layers of the entorhinal cortex (EC) at varying levels at P10, but exhibits a drastic drop in expression levels at P30. Tenm2, Tenm3, and Tenm4 were similarly enriched in the s.p. layer but were also present in other layers such as the stratum oriens (s.o.), radiatum (s.r.), and lacunosum moleculare (s.l.m.) at P0 and P10. However, Tenm2, Tenm3, and Tenm4 became mostly restricted to the s.p. layer at P30 and diminished in the EC layers. All Tenms were highly enriched in the stratum granulare (s.g.) layer of the DG with

*Experimental Strategy*

**Fig 1. Experimental strategy.** Schematic depiction of the experimental strategy to reveal Tenm and Lphn expression map in dorsal-ventral and lateral-medial axis of mouse hippocampus through RNA in situ hybridization in multiple horizontal and sagittal sections at different developmental time points. (1) Four anatomical sections from sagittal and horizontal orientations of brain obtained from 4 developmental stages. (2) Single molecule RNA in situ hybridization with Tenm1, 2, 3, 4 and Lphn1, 2, 3 and Flrt1, Flrt2, Flrt3 probes (performed at P30 only). (3) Automated imaging of slides with RNA in situ hybridized brain sections. (4) Manual image segmentation in accordance with Allen Brain Atlas reference images for sagittal sections and quantification using threshold algorithm. Note that the horizontal sections were not used for quantifications in this study.

reduced levels of Tenm2 and Tenm4 at P30 (Figs 2E–2G and S6–S9). Thus, in pyramidal neurons and granule cells of the hippocampus, multiple teneurins are co-expressed in the same neurons with peak expression levels correlating with peak synaptogenesis stages.

We next analyzed the expression levels of Tenms in individual lateromedial sections across different ages to further investigate their spatiotemporal expression patterns. Within the s.p. layer, Tenm1 showed increased expression towards the medial axis of the s.p. layer at P10 with a slightly opposite shift (towards lateral) at P30 (Figs 3A, 3E, and S2–S9). Compared to this, the expression level was much higher in the s.g. layer of DG, with a clear lateral to medial gradient. Tenm2 expression showed a striking lateromedial gradient at the E16.5 stage across all hippocampal regions. The expression became confined to the subiculum at P0 and vanished at P10 and P30 in this region. Like Tenm1, Tenm2 expression also displayed specificity in the s.p. layer with high levels at postnatal stages. EC layer 2 (L2) was particularly enriched in Tenm2 expression at P10 (Figs 3B, 3E, and S2–S8). Tenm3 was expressed at high levels in the CA region at the embryonic stage, decreased at P0 and P10, and peaked again at P30 with stable lateromedial expression in the s.p. layer. Contrary to the CA region, Tenm3 expression in the subiculum was enriched at P0 and showed lower levels at P10 and P30. Tenm3 expression in the EC layers was lower when compared to the s.p. and s.g. layers (Figs 3C, 3E, and S2–S9). Tenm4 levels in the CA region were much lower at the embryonic stage and detected at a higher level in CA1 at P0, later confining to the s.p. layer during the P10 and P30 stages. Expression in the DG was higher with a clear lateromedial gradient during the embryonic stage compared to postnatal stages, when the expression was specific to the s.g. layer in a gradient fashion. In the subiculum and EC layers, the enrichment was higher in the embryonic stage when compared to P10 and P30, where the higher levels were detected in L5/6 (Figs 3D, 3E, and S2–S9).

We next analyzed Lphn expression in a similar fashion. At the embryonic and P0 stages, all Lphns were expressed at similar levels in the hippocampal region with the exception that Lphn2 levels were slightly lower in CA3 and DG (Figs 4A–4D, S10, and S11). At postnatal stages, all Lphns showed similar expression levels in the s.o. and s.p. layers of CA1. However,

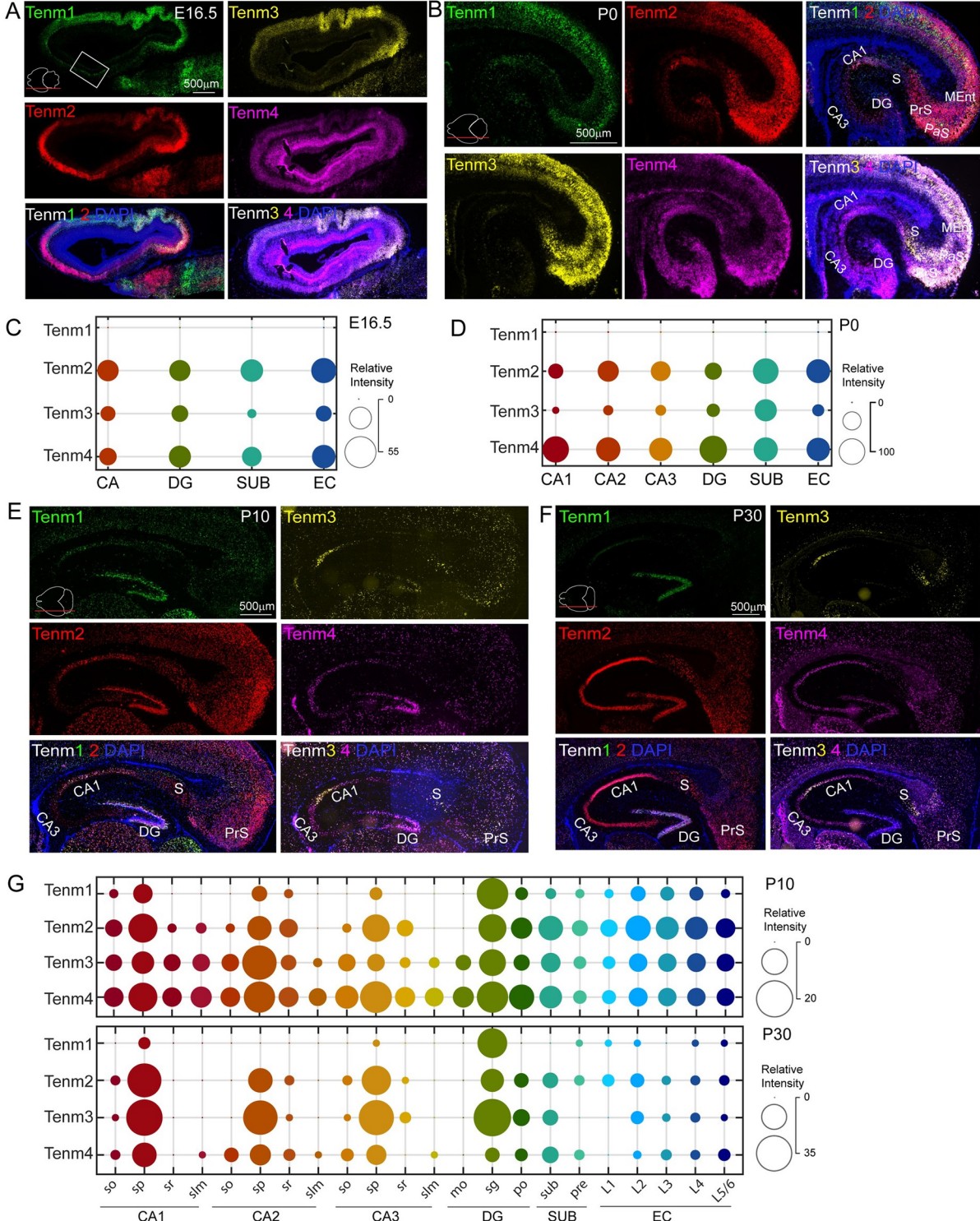

**Fig 2. Teneurin expression is developmentally regulated within the hippocampal regions.** (A) Sagittal section of E16.5 mouse brain labeled with Tenm1, Tenm2, Tenm3, and Tenm4 in situ hybridization probes. Note that the region corresponding to the embryonic hippocampus is marked as a rectangle box. The removal of the embryonic brain out of the skull and embedding within the cryosectioning solution flattened the brain and altered its original posture; hence, the hippocampus appears as an anterior structure. (B) Close-up images of hippocampal region of P0 brain labeled with all teneurins show their dynamic expression. (C, D) Bubble plots show the expression levels of all teneurins in E16.5 and P0 hippocampal subregions. (E) P10 and (F) P30 hippocampus sagittal sections reveal the expression pattern of all teneurins. (G) Bubble plots show expression levels of all teneurins in P10 and P30 hippocampal subregions and layers. Images in panels A, B, E, and F are the

enlarged and annotated version of selected corresponding images from S2, S4, S6, and S8 Figs. so, stratum oriens; sp, stratum pyramidale; sr, stratum radiatum; slm, stratum lacunosum-moleculare; mo, molecular layer; sg, stratum granulare; po, polymorph layer; S, SUB, subiculum; PrS, pre, presubiculum; L1–L6, layer 1–6; PaS, parasubiculum; CA1, CA3, cornu Ammonis 1, 3; DG, dentate gyrus; EC, entorhinal cortex; MEnt, medial entorhinal cortex.

in the CA2 and CA3 s.p. layer, Lphn1 and Lphn3 were predominant, with Lphn2 being virtually undetectable in the CA3 s.p. layer at P10. The same trend was observed in the s.g. layer of DG. Lphn2 expression was mostly in the subiculum and pre-subiculum, whereas Lphn1 and Lphn3 were distributed throughout the subiculum albeit with a slight decrease in the pre-subiculum at P30. All layers of EC (except L1) showed essentially uniform distribution of all Lphns; however, at P30 they dropped their expression levels uniformly throughout all EC layers (Figs 4E–4G, S12, and S13).

Lateromedial expression analyses of Lphns revealed that all Lphns are expressed from embryonic to postnatal stages. Lphn1 expression in the embryonic stage showed a clear gradient in the CA, DG, and EC regions, which was also evident at P0. However, distribution in the DG across the lateromedial axis reversed at P0. Expression in the subiculum was irregular across the lateromedial axis. At postnatal stages, Lphn1 expression was confined to the s.p. layer of CA1, CA2, and CA3, and the s.g. layer of DG without any significant gradient. All subiculum regions and EC layers showed lower expression levels than the s.p. and s.g. layers (Figs 5A, 5D, and S10–S13). Lphn2 expression in the embryonic EC showed a mediolateral (high towards medial, low towards lateral) pattern. In the CA, DG, and subiculum regions, there was no obvious gradient in expression. P0 expression in CA1 exhibited a lateromedial gradient and apparent specificity to the lateral-most region in CA2. CA3, however, did not show any expression of Lphn2. High enrichment of Lphn2 was found in the subiculum region when compared to the CA, DG, and EC regions. At P10 and P30, the expression became mostly specific to the s.p. layer of CA1. The CA2 s.p. layer had a much lower level of expression consistent with the P0 stage. Lphn2 expression in the subiculum was lower compared to CA1; however, the levels were much higher in the pre-subiculum region at P10 compared to P30, where it was virtually undetectable (Figs 5B, 5D, S12 and S13). Embryonic expression of Lphn3 was similar to Lphn2, with high enrichment observed in the EC with an obvious lateromedial gradient, which was reflected in the P0 stage as well. Lower and gradient expression in the CA region with the same pattern in P0 was also observed. Subiculum expression was irregular across the lateromedial axis, but DG showed a smooth gradient in Lphn3 expression. P0 CA1 had abundant Lphn3 expression as gradient and laterally restricted expression in CA2 and CA3 with undetectable levels in the DG. The subiculum followed embryonic expression patterns with irregularity along the axis, but at much higher levels. Unlike Lphn2, postnatal expression of Lphn3 was observed in the s.p. layer of CA1, CA2, and CA3, and the s.g. layer of DG with higher levels in the CA1 s.p. layer. There was no gradient of Lphn3 in the subiculum and EC layers (Figs 5C, S12 and S13). Lphn expression patterns in the dorsoventral axis visualized though the horizontal plane were mostly similar to that of the lateromedial axis (Figs 6, S11 and S13). With horizontal sections, we were able to identify EC regions from 2 ventral sections. In P30, the ventral-most EC (caudomedial entorhinal cortex) regions showed striking enrichment of Lphn1 in upper (L-II), Lphn2 in mid (L-III, L-IV), and Lphn3 in lower (L-V/VI) layers (Fig 6A–6D).

Our observation of Tenms and Lphns expression in the CA1 s.p. layer and DG m.o layer (Figs 2E and 4G) does not rule out if they are expressed in the same cells of these layers. To find out, we performed combinatorial in situ hybridization using various Tenms and Lphns probes (Fig 7A). We found Tenms and Lphns expression largely overlapped in CA1 s.p. cells

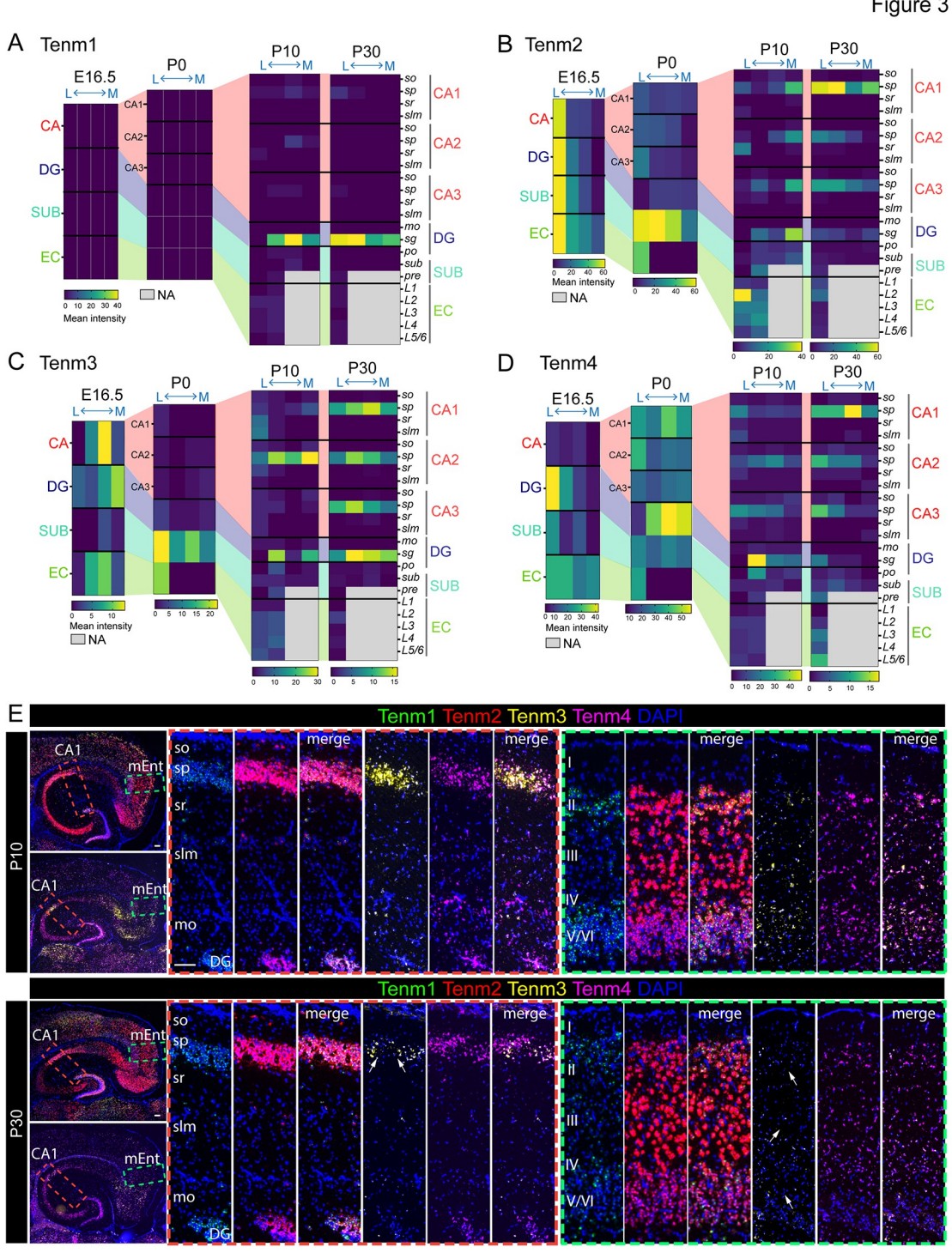

**Fig 3. Teneurin expression is spatiotemporally defined.** (A–D) Heat maps show latero-medial expression levels of teneurins across different ages in the hippocampal subregions and layers. (E) Horizontal sections of hippocampus of P10 and P30 mice show layer-specific teneurin expression patterns. Note the striking difference in expression levels of Tenm3 between P10 and P30 in the CA1 s.p. layer and all EC layers (white arrows). Images in panel E are the enlarged and annotated version of selected corresponding images from S7 and S9 Figs. so, stratum oriens; sp, stratum pyramidale; sr, stratum radiatum; slm, stratum lacunosum-moleculare; mo, molecular layer; I–VI, layer 1–6; CA1, cornu Ammonis 1; DG, dentate gyrus; MEnt, medial entorhinal cortex. The data underlying heatmaps in A–D can be found in S1 Data. Scale bar, 100 μm.

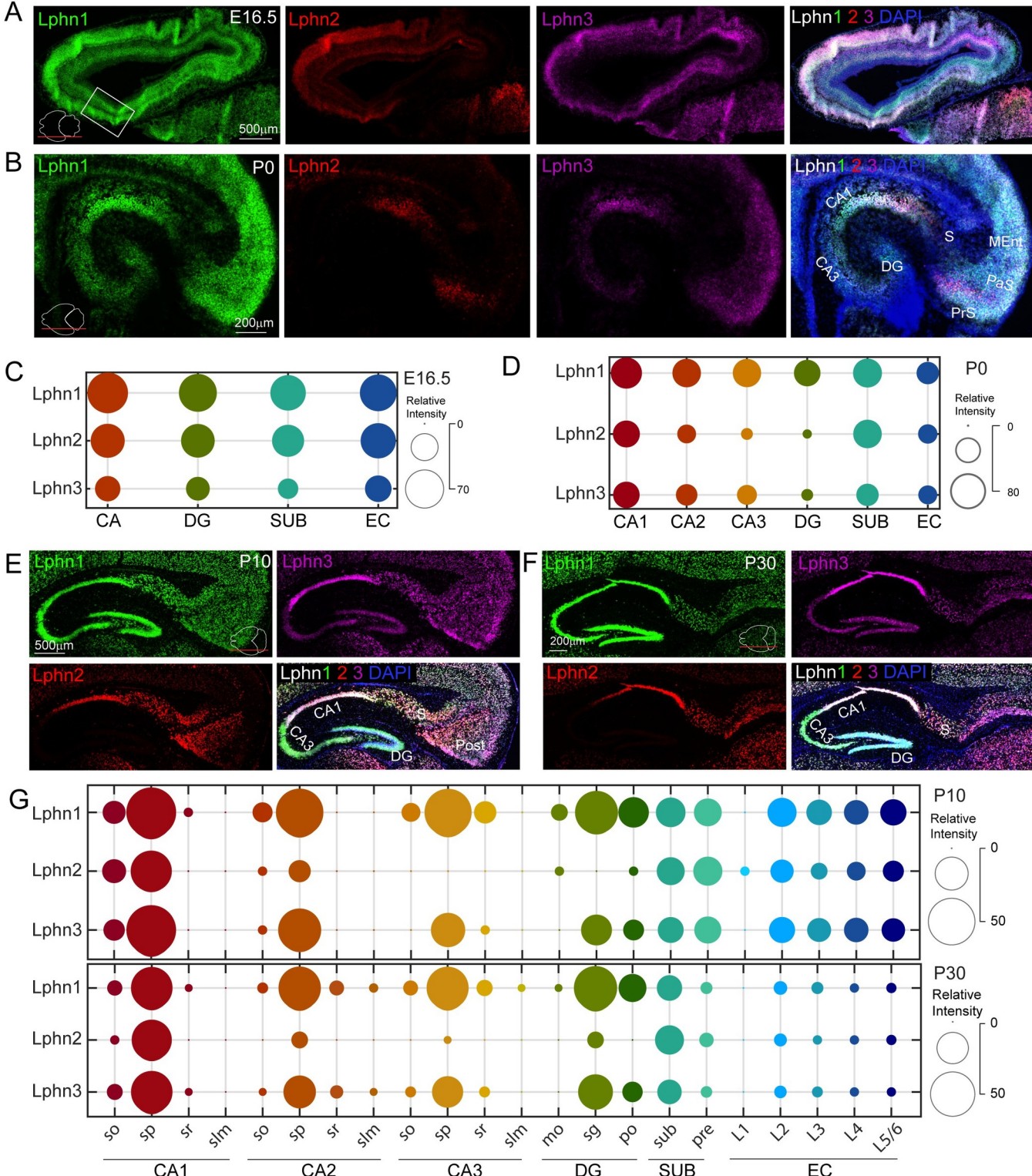

**Fig 4. Latrophilin expression is developmentally regulated within the hippocampal subregions.** (A) Sagittal section of E16.5 mouse brain labeled with Lphn1, Lphn2, and Lphn3 in situ hybridization probes. Note that the region corresponding to the embryonic hippocampus is marked as a rectangle box. The removal of the embryonic brain out of the skull and embedding within the cryosectioning solution flattened the brain and altered its original posture; hence, the hippocampus appears as an anterior structure. (B) Close-up images of hippocampal region of P0 brain labeled with all latrophilins show their dynamic expression. (C, D) Bubble plots show the expression levels of latrophilins in E16.5 and P0 hippocampal subregions. (E) P10 and (F) P30 hippocampus sagittal

sections reveal the expression pattern of latrophilins. (G) Bubble plots show expression levels of all latrophilins in P10 and P30 hippocampal subregions and layers. Images in panel A, B, E, and F are the enlarged and annotated versions of selected corresponding images from S10 and S12 Figs. so, stratum oriens; sp, stratum pyramidale; sr, stratum radiatum; slm, stratum lacunosum-moleculare; mo, molecular layer; sg, stratum granulare; po, polymorph layer; S, SUB, subiculum; PrS, pre, presubiculum; L1–L6, layer 1–6; PaS, parasubiculum; Post, postsubiculum; CA1, CA3, cornu Ammonis 1, 3; DG, dentate gyurs; EC, entorhinal cortex; MEnt, medial entorhinal cortex.

and DG granule cells, albeit with varying levels of expression (Fig 7B–7F). However, in the DG polymorph layer (p.o.), the excitatory hilar cells showed both overlapping and nonoverlapping patterns of Tenm and Lphns expression. For example, in these cells Tenm3 and Lphn3 expression does not overlap (Fig 7B), whereas Lphn1 and Tenm4 expression largely does (Fig 7F).

We next asked if the lateromedial heterogeneity of Tenm and Lphn corresponds to or correlates with the dorsoventral expression pattern (medial to dorsal and lateral to ventral) through RNA-seq data (Fig 8D). By analyzing Hipposeq RNA-Seq data [33], we found Tenm and Lphn expression in many hippocampal regions correlated with our observations. Dorsoventral expression of Tenm1 in CA1 correlated with lateromedial expression patterns in the CA1 and CA3 s.p. layer at P30 (Figs 3A and 8A). High enrichment of Tenm4 in vDG (ventral DG) reflected as high levels in the lateral axis at P30 (Figs 3D, 8A and S9). There were no striking expression gradients of Lphns in dorsoventral data, which reflects our observations in postnatal expression. Prominent enrichment of Lphn2 in both dCA1 (dorsal CA1) and vCA1 validates our lateromedial results (Figs 5B, 8B and S13). It should be noted that Tenm2 expression in DG and Tenm4 in CA1 from our lateromedial data does not correlate with dorsoventral Hipposeq data. This could be due to cell type-specific Cre lines used to target cell populations in DG and CA1 in the Hipposeq study [33]. In this lateromedial RNA-seq and subsequent bulk and single-cell RNA-seq (scRNA) analysis, we also included the Flrts (Flrt1-3, fibronectin leucine-rich transmembrane) due to their known interaction with Lphn and Tenm and also to investigate if they display any spatial expression patterns [4]. Flrt1 expression in CA1 and CA3 showed a dorsoventral contrast, whereas Flrt2 and Flrt3 expression was more-or-less the same across this axis (Fig 8C). To confirm Flrts expression in situ, we performed ISH with Flrt1, Flrt2, and Flrt3 probes. Dorsoventral gradient expression of Flrt1 and Flrt2 in CA1 inferred from Hipposeq data largely correlated with our lateromedial in situ data. Flrt3 expression largely remained uniform in the dorsoventral and lateromedial axes, albeit with a slight gradient in the CA1 and CA3 s.p. layers (Fig 8E and 8F).

The mammalian hippocampus is known to contain heterogeneous cell populations and even within-cell-type heterogeneity [30,34]. In our in situ mapping, we found Tenm and Lphn expression was mostly restricted to the s.p. and s.g. layers, which are known to encompass excitatory and other cell types. To identify the cell type-specific expression of Tenm and Lphn in the context of our spatiotemporal map and relative Flrt expression, we analyzed subclass-specific single cell RNA-seq (scRNA-seq) datasets (whole cortex and hippocampus) from the Allen Brain Map [35]. Within the glutamatergic class of neurons, Tenm2, Tenm4, Lphn1, and Lphn3 show generally uniform expression (S14 Fig). Tenm1, Tenm3, and Lphn2 are expressed in a subset of cell populations which includes entorhinal (ENT) and intratelencephalic (IT) layers as well as pyramidal track (PT) neighborhoods. Flrt1 expression is high in the PT neighborhoods, while Flrt2 showed patchy expression across glutamatergic clusters. Flrt3 expression was enriched in CA3, entorhinal (ENT), and medial entorhinal (mENT) subclusters. In the GABAergic class, Tenm1, Tenm2, Tenm4, Lphn1, Lphn3, and Flrt2 expressed in similar levels to the glutamatergic class. Tenm3 is expressed in subclasses of GABAergic neurons such as Sst and PV. Lphn2 and Flrt1 showed much more narrowed expression within Sst and Sst-Chodl clusters. As for non-neuronal cell types, only Lphn3 is expressed in oligodendrocyte and astrocyte subtypes and Flrt2 in cells of the immune and vascular systems (S14 Fig).

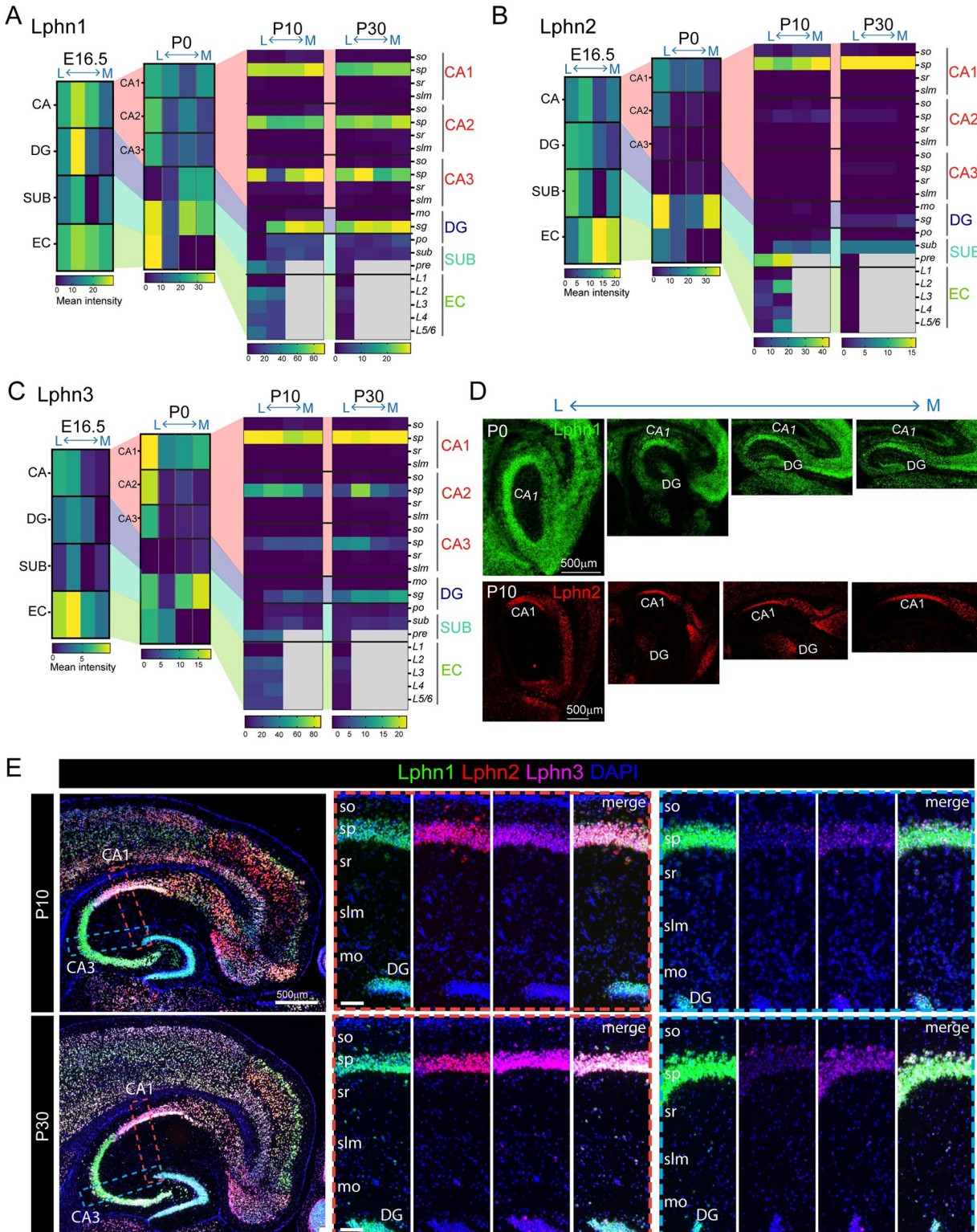

**Fig 5. Latrophilins display spatiotemporal expression patterns.** (A–C) Heat maps show latero-medial expression levels of latrophilins across different ages in the hippocampal subregions and layers. (D) Latero-medial sagittal sections of hippocampus of P0 and P10 mice show latrophilins expression patterns in hippocampal subregions. (E) Horizontal sections of hippocampus of P10 and P30 mice show layer-specific latrophilin expression patterns in CA1 and CA3. Images in panels D and E are the enlarged and annotated version of selected corresponding images from S10, S12 and S13 Figs. The data underlying heatmaps in A–C can be found in S1 Data. so, stratum oriens; sp, stratum pyramidale;

sr, stratum radiatum; slm, stratum lacunosum-moleculare; mo, molecular layer; sg, stratum granulare; po, polymorph layer; SUB, subiculum; pre, presubiculum; L1–L6, layer 1–6; CA1, CA3, cornu Ammonis 1, 3; DG, dentate gyurs; EC, entorhinal cortex.

To further probe deeper into the expression patterns in neuronal, astroglial, and hippocampal projection classes, we analyzed mouse cell types and tissues from bulk RNA-Seq data through the ASCOT (alternative splicing catalog of the transcriptome) database, which includes both gene expression and alternative splicing results [36]. The dorsoventral data sets (Fig 8A–8D) were analyzed in ASCOT datasets independently and they largely match (Fig 9A). In the dCA1 PoS (dorsal CA1 post subiculum) projection class, Tenm2, Tenm3, and Lphn3 expression levels were high and Lphn1 and Flrt2 were lower. Tenm1, Tenm4, Flrt1, and Flrt3 were virtually undetectable. In the vCA1 Amygdala projection class, high enrichment of Tenm1, Tenm2, and Lphn1 was observed. Tenm3, Tenm4, and Lphn3 were lower in expression. Lphn2 and all Flrts were not expressed in this projection class. In vCA1 Nucleus accumbens (NAc)-projecting neurons, Lphn1 and Tenm2 were highly enriched with moderate levels of Tenm1, Tenm3, Tenm4, Lphn3, and Flrt2 expression. Lphn2 and Flrt1 expression was negligibly low (Fig 9A). Further independent analysis on hippocampal subclass (neuronal and non-neuronal) expression data showed higher expression of all Tenms and Lphn1 in Gad2 + interneurons. In Nxph3+ subiculum-entorhinal cells, Tenm2 and Lphn1 levels were higher. In the Slc17a6+ excitatory subclass, Tenm2 levels were much higher compared to all other subclass cells. Overall, non-neuronal cells showed decreased expression of all Tenms, Lphns, and Flrts, except Flrt2 expression levels were significantly high in Dcn+ fibroblast-like cells (Fig 9B). Analysis of Tenms, Lphns, and Flrts in hippocampus clusters through single-nuclei RNA-seq (sNuc-seq) [37] painted a similar picture compared to all other bulk, scRNA-seq datasets and our in situ analysis (Fig 9C). However, further analysis of hippocampus subcluster RNA-seq revealed cell type-specific expression of Tenms, Lphns, and Flrts. For example, Tenm3 expression was high in GABAergic clusters, whereas subcluster data showed Tenm3 is not expressed in cells that express serotonin receptors (Gad2_Htr3a_3 subcluster). In Glial clusters Lphn1 expression was high, but subcluster analysis revealed that microglia express Lphn1 at a much lower level. Lphn1 enrichment is mainly in *oligodendrocyte* precursor cells (OPC) and astrocytes (Fig 9D).

We next asked if the expression of Tenms, Lphns, and Flrts as inferred through in situ and RNA-seq analysis reflects on mRNA that undergo translation (translatome). We analyzed the ribosome-engaged transcriptome datasets (Splicecode database) for 2 excitatory: Camk2a + and Grik4+ and 4 inhibitory: Sst+, Scnn1a+, Pvalb+, and Vip+ classes in the hippocampus and cortex (S15A Fig) [38]. These analyses showed similar expression patterns of Tenms, Lphns, and Flrts in inhibitory and excitatory classes when compared to our in situ and RNA-seq datasets (S15B–S15D Fig).

To analyze Tenm, Lphn, and Flrt expression outside the brain, we used ASCOT datasets and found these genes are indeed similarly enriched in many other organs, with Lphn2 being represented in the cells of the musculoskeletal, immune, digestive, and reproductive systems, with the exception that Tenm1 is not detected in any non-neural tissues. Tenm4 is highly enriched in ovarian cells and Flrt3 in bone marrow dendritic cells. Lphn2 shows higher expression in liver endothelial cells (S16 Fig). These expression patterns strongly suggest possible roles of these genes outside the CNS.

## Discussion

In mapping the spatiotemporal expression patterns of Tenms and Lphns in the hippocampus, we here describe a remarkable heterogeneity in the developmental trajectories of their

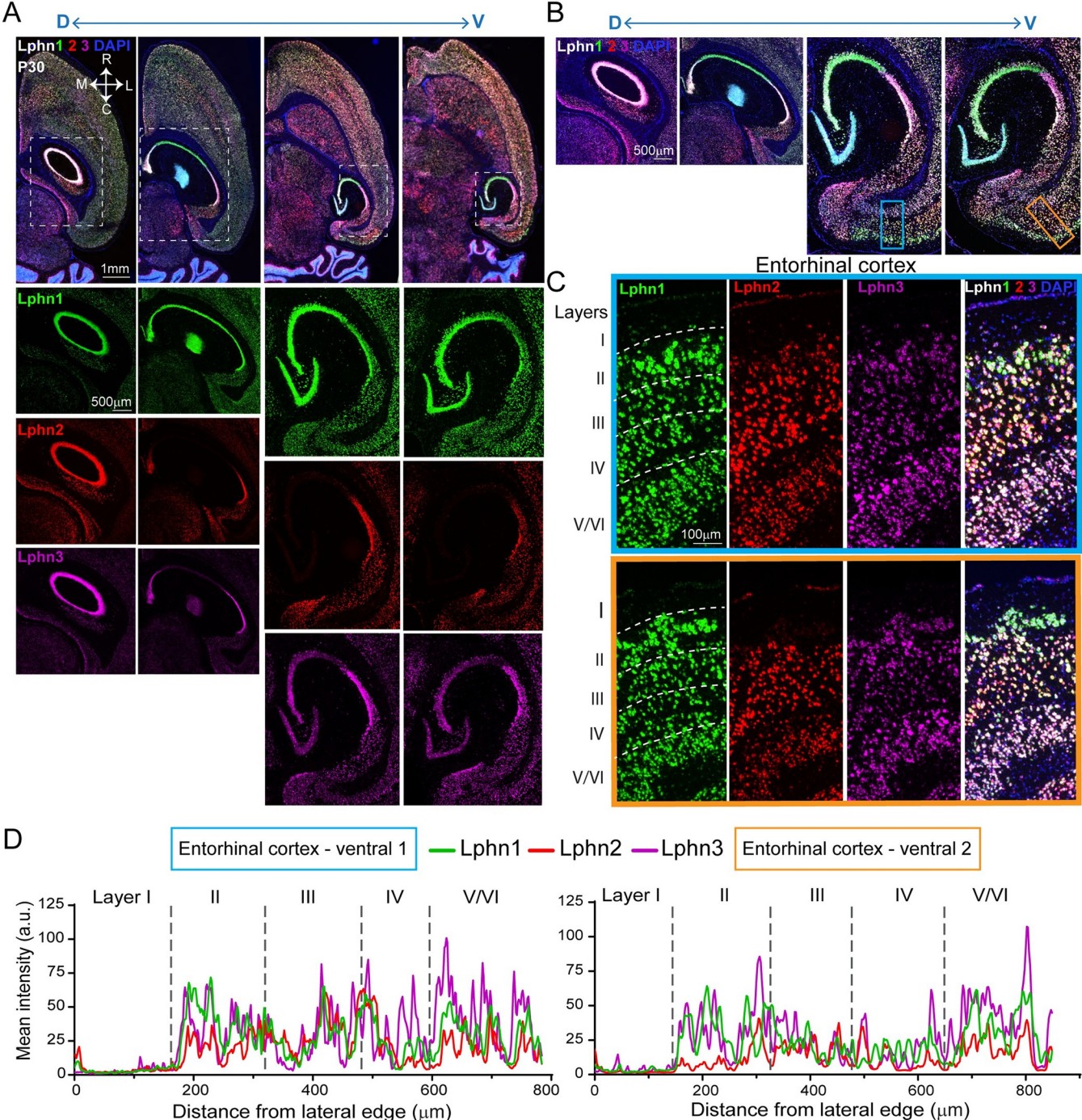

**Fig 6. Latrophilin expression is relatively uniform across hippocampal dorsoventral axis.** (A) Microscopic images show expression of Lphn1, Lphn2, and Lphn3 across dorsoventral axis of brain hemisphere. Hippocampal regions (dotted line box) are enlarged below and shown with each Lphn expression. (B) Merged Lphns labeling to show dorsoventral hippocampal and EC regions. (C) Enlarged images of EC regions (blue and boxes in B) to reveal Lphn expression pattern in individual EC layers. (D) Quantification of mean fluorescence intensity in ventral 1 and 2 sections (blue and orange boxes in B). Images in panels A and B are the enlarged versions of selected corresponding images from S13 Fig. Note that the third image in panel B is a rotated version of the first image in panel E (P30) in Fig 5, re-used to show entorhinal cortex. The data underlying graphs in D can be found in S1 Data.

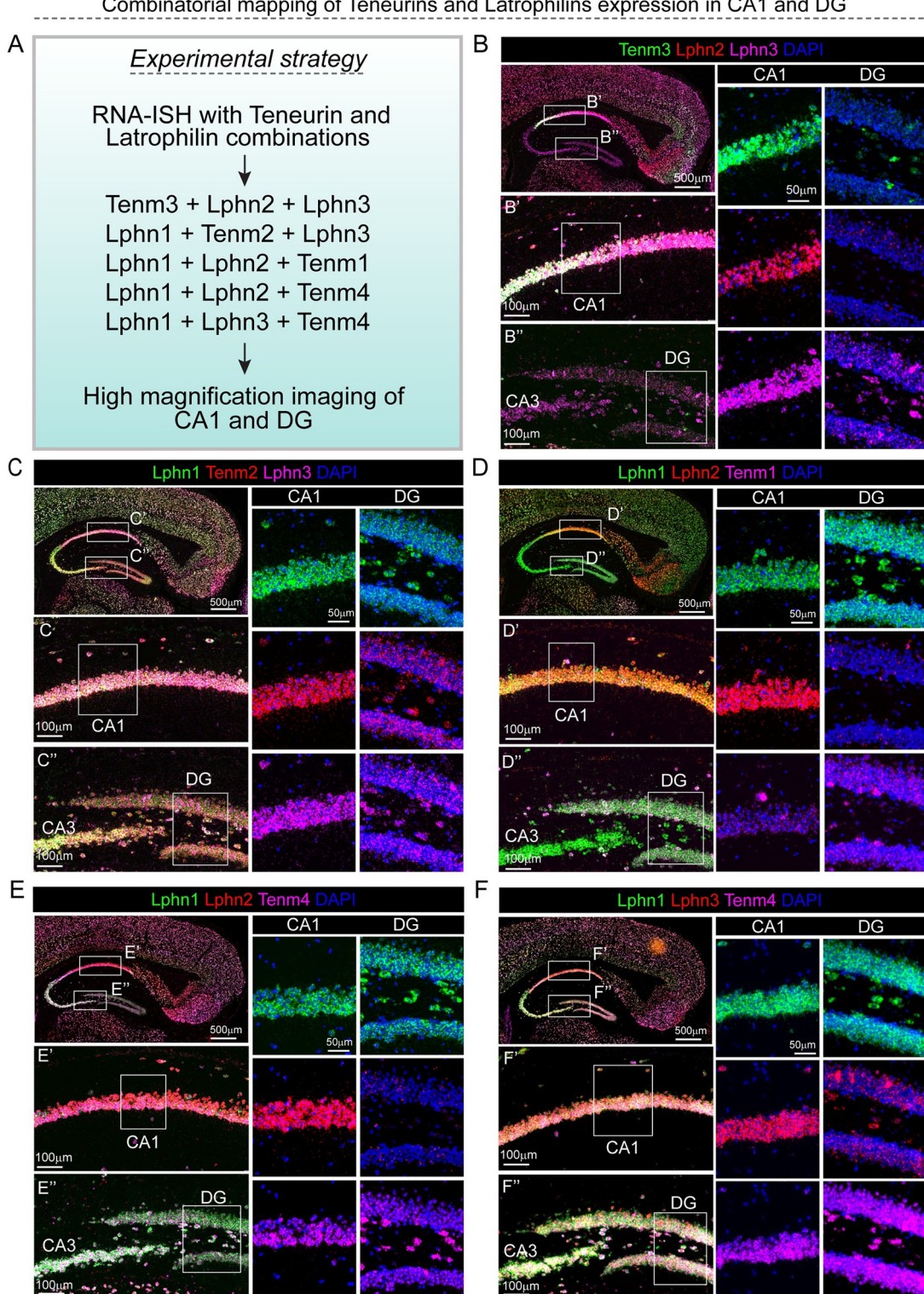

**Fig 7. Expression of teneurins and latrophilins largely overlap in the CA1.** (A) Experimental strategy for Tenm and Lphn RNA-FISH combinations. (B–F) RNA-FISH images show different combinations of Tenm and Lphn expression in CA1 and DG regions (boxes in top left panel marked with alphabet' and alphabet" and their corresponding high magnifications below). Further magnifications of CA1 and DG regions provided on the right side within each panel. CA1, CA3, cornu Ammonis 1, 3; DG, dentate gyurs.

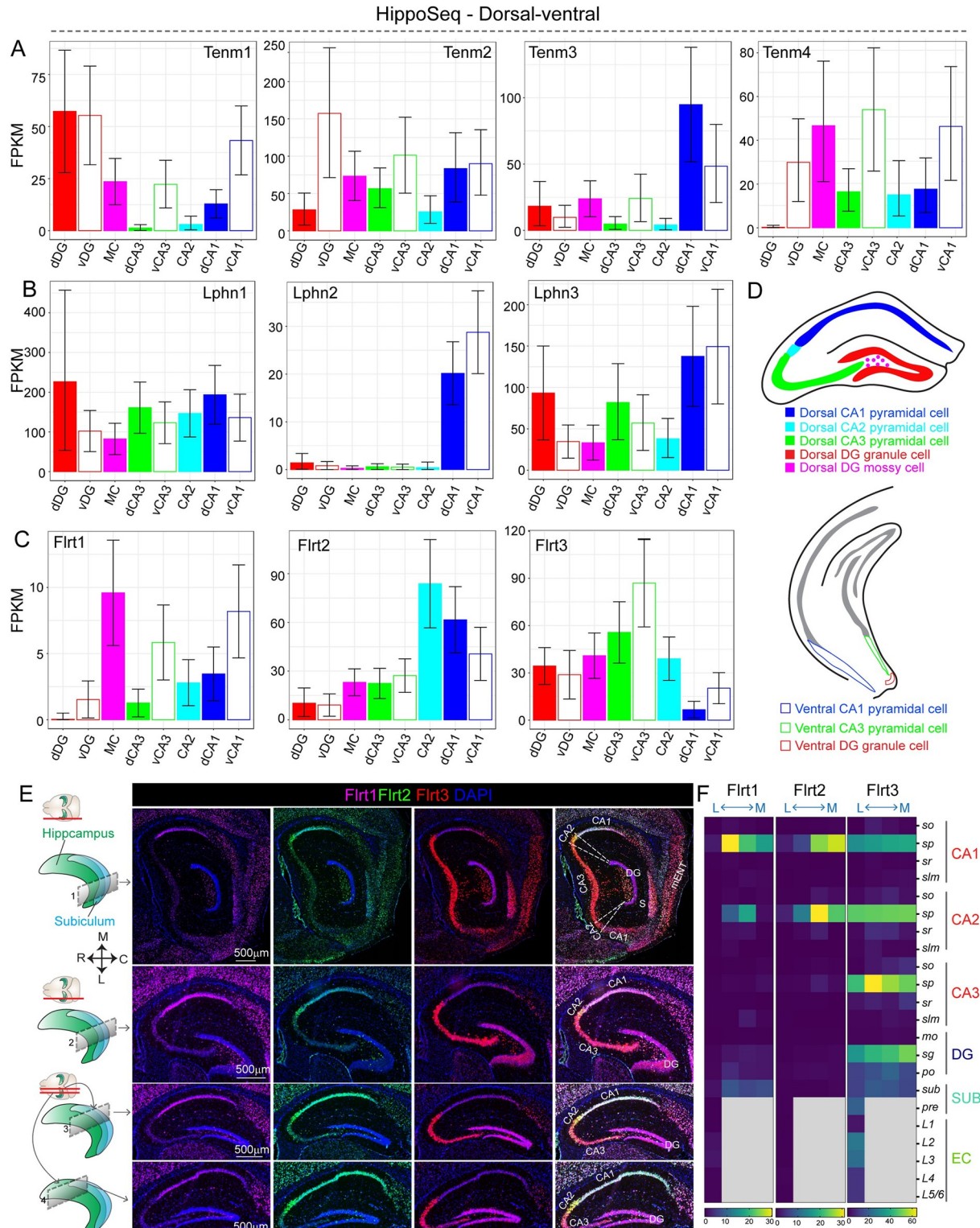

**Fig 8. Mediolateral spatial expression patterns of teneurin and latrophilins recapitulated through dorsoventral RNA-seq expression analysis.** (A) Bar graphs show all teneurin expression at RNA level in dorso-ventral axis of the hippocampus. Note the striking difference in Tenm1 expression in dCA3 vs. vCA3 and Tenm4 expression in dDG vs. vDG and dCA3 vs. vCA3. (B) Bar graphs show RNA expression levels of Lphns in dorso-ventral axis. Note the enrichment of Lphn2 in the CA1 region. (C) Bar graphs show expression levels of Flrts in dorso-ventral axis of the hippocampus. (D) Schematic depicting the dorsal and ventral hippocampus and its subregions from which the RNA-seq data was

obtained. (E) Microscopic images show RNA-FISH for Flrt1, Flrt2, and Flrt3 across mediolateral axis of hippocampus at P30. (F) Heatmap shows region-specific intensity values for Flrt1, Flrt2, and Flrt3. Note the striking gradient of Flrt2 in CA1 and CA2 s.p. layer. RNA-seq data (in panels A, B, and C) obtained from Hippocampus RNA-seq atlas (https://hipposeq.janelia.org/). Schematic diagrams in D are modified from images in Hippocampus RNA-seq atlas. The data underlying graphs in A–C and heatmaps in F can be found in S1 Data. dDG, vDG, dorsal and ventral dentate gyrus; MC, Mossy cell; dCA3, vCA3, dorsal and ventral CA3 pyramidal cell; dCA1, vCA1, dorsal and ventral CA1; S, subiculum; mEnt, medial entorhinal cortex, so, stratum oriens; sp, stratum pyramidale; sr, stratum radiatum; slm, stratum lacunosum-moleculare; mo, molecular layer; sg, stratum granulare; po, polymorph layer; SUB, subiculum; pre, presubiculum; L1–L6, layer 1–6; CA1, CA3, cornu Ammonis 1, 3; DG, dentate gyurs; EC, entorhinal cortex; FPKM, Fragments Per Kilobase of transcript per Million mapped reads.

expression levels. By integrating RNA in situ analyses of Tenms and Lphns using sagittal and horizontal section planes with existing RNA-seq datasets, we show that Tenms and Lphns exhibit striking developmental dynamics with postnatal regionalization to CA pyramidal cell layers and the DG GCL. One particularly notable finding was how *Tenm1* is only detected after P10, suggesting a specifically postnatal role. All expression of Tenms and Lphns displays a lateromedial gradient at some level across the hippocampus. The dorsal and ventral hippocampus pole-restricting expression of Tenms, Lphns, and Flrts in the CA1 projection class neurons suggests a functional dichotomy of their spatial expression [39]. This hypothesis can be tested by selective manipulation of these genes in their site or cell populations of expression and assessing synaptic plasticity and performing circuit-dependent behavioral tests. Within the hippocampal glial subclusters, all Tenms, Lphns, and Flrts are expressed at varying levels, with Lphn1 being high in OPC and astrocytes, suggesting additional non-neuronal functions for these genes. From an evolutionary point of view, the roles of Tenm and Lphn in hippocampal circuit development might be relevant to the emergence and conservation of hippocampal–parahippocampal connectivity [30]. Selective expression of specific Tenm, Lphn, and Flrt paralogs outside of the CNS suggests they may play similar cell-adhesion roles in relevant tissues [40–42]. Given that these genes are also alternatively spliced, it is possible that the non-neural tissues express functionally defined tissue-specific isoforms.

Several previous studies have examined the spatial topography of gene expression in the hippocampus, primarily through RNA in situ analysis and more recently with RNA-seq approaches [33,34,43]. The CA3 alone has 9 neighborhoods that are spatially distinct in gene expression across the dorsoventral and proximodistal axes [44]. Mounting evidence suggests that the classical CA regions, subiculum pyramidal neurons, and dentate granule cells all exhibit prominent spatial within-cell heterogeneity. Importantly, the differential expression of neuronally relevant genes across the hippocampus is likely to underlie heterogeneity in higher-order structure and function [45–49]. There are multiple ways by which spatiotemporal gene expression could regulate neuronal function. For example, the dentate gyrus is characterized by the expression of genes involved in the formation of new neurons, while the CA1 region is characterized by the expression of genes involved in synaptic plasticity and memory consolidation [50]. The spatial segregation of these gene expression profiles is thought to contribute to the formation of functional circuits that support different aspects of learning and memory. Another way that spatial topography of gene expression in the hippocampus contributes to neuronal function is through the regulation of synaptic plasticity, which is critical for learning and memory. Many of the genes that are differentially expressed in different hippocampal sub-regions are involved in the regulation of synaptic plasticity. The spatial organization of hippocampal gene expression may allow for the precise regulation of synaptic plasticity in different subregions in response to different inputs. Interestingly, we also observed that Tenm1 expression in the CA1 region coincides with the onset of the speculated critical period (P14–P24) for experience-dependent plasticity in this region ([51–54]), suggesting Tenm1 may play a role in regulating synaptic plasticity during this period.

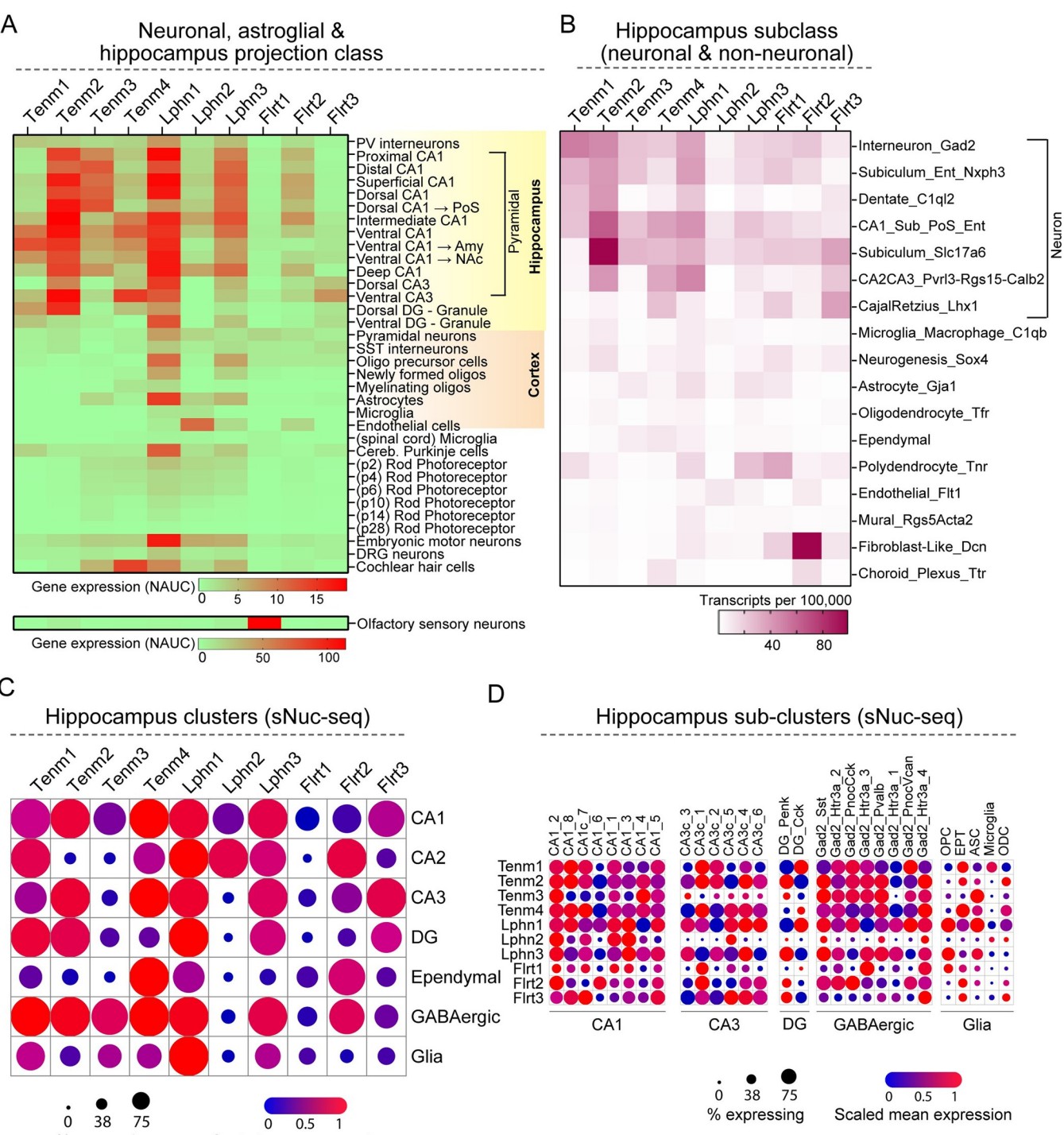

**Fig 9. Teneurin and latrophilin are specifically enriched in neuronal and glial subclass.** (A) Heat map shows expression of Tenm, Lphn, and Flrts in hippocampal projection class neurons, cortical neurons, and non-neuronal cell types. (B) Heat map shows expression of Tenm, Lphn, and Flrts in hippocampal subclass (neuronal and non-neuronal) cells based on specific marker expression in each subclass. (C) Bubble plot combined heat map shows expression and percent of cells expressing Tenm, Lphn, and Flrts in neuronal and non-neuronal cell clusters of hippocampus. (D) Bubble plot combined heat map shows expression and percent of cells expressing Tenm, Lphn, and Flrts in hippocampal subclusters. Bulk and scRNA-seq data obtained from ASCOT database (http://ascot.cs.jhu.edu/) (A), Dropviz database (http://dropviz.org/) (B), and single nucleus RNA-seq (sNuc-Seq) of adult mouse hippocampus (https://singlecell.broadinstitute.org/single_cell/study/SCP1/-single-nucleus-rna-seq-of-cell-diversity-in-the-adult-mouse-hippocampus-snuc-seq) (C, D). NUAC (Normalized Area Under the Curve). CA1, CA2, CA3, cornu Ammonis 1, 2, 3; DG, dentate gyurs; OPC, *oligodendrocyte* precursor cells; EPT, ependymal; ASC, astrocytes; ODC, oligodendrocytes.

Through topographic mapping, Tenm3 has been shown to be expressed in multiple interconnected areas of the hippocampus. Furthermore, this selective topography of Tenm3, along with repulsive expression of Lphn2 was shown to guide the assembly of CA1 subiculum connections in the medial and lateral hippocampal networks [18,32]. The expression map of Lphn2 and Tenm3 shown in these studies largely correlates with the expression patterns and gradients found in our study. Additionally, our results on lateromedial gradient expression (especially in CA1) of Flrt1 and Flrt2 might also suggest their similar role in connectivity-specific function in the hippocampus. Higher levels of Tenm2 in CA regions are persistent in P0 and P30 but diminish in the subiculum and EC layers at P30. This might indicate its selective role in connections involving CA regions. Although the function of Tenm2 is not fully explored, a few studies show its interaction with Lphns. Through biochemical assays, transynaptic complex mediated by Tenm2 and Lphn1 interaction in the rat hippocampus was shown [5]. Recently, Tenm2 was shown to form a trimeric complex with Lphn3 and Flrt3, indicating possible multifaceted interaction of Tenm2 to mediate synaptic function [55]. Tenm4 was shown to be crucial in guiding axon projection and regulating oligodendrocyte differentiation [56,57]. However, Tenm4's role in hippocampal networks remains unexplored. Our results on Tenm4 gradient expression in the CA and DG regions provide valuable insights for future investigations.

The Allen Mouse Brain Atlas provides a whole-genome resource to understand the general spatial organization of the hippocampus and other brain regions [43]. However, Allen's Developing Mouse Brain Atlas does not include in situ data for Tenm and Lphn expression (except Tenm3) [58]. Moreover, our dorsoventral spatiotemporal map of Tenms and Lphns complements not only the lateromedial map but also existing RNA-seq atlases. Therefore, our study offers a valuable hypothesis-generating resource for future investigations to link Tenm and Lphn topography to function. In conclusion, our study provides a comprehensive spatiotemporal map of Tenm and Lphn expression in the hippocampus, revealing their heterogeneous expression patterns across developmental and spatial axes.

## Methods

### Mice

Animal experiments conducted in this study were in accordance with National Institute of Health Guidelines for the Care and Use of Laboratory Mice and protocols (No. 18846) approved by the Stanford University Administrative Panel on Laboratory Animal Care (SUAPLAC)/Institutional Animal Care and Use Committee (IACUC). Wild-type mice used in this study were outbred Crl: CD1(ICR) mice, obtained from Charles River Laboratories.

### Mouse brain tissue sampling and preparation

To harvest E16.5 developmental stage brains, a timed mating with a WT male and virgin WT female (approximately 10 weeks old) was set up. E0.5 was defined as when the vaginal plug was identified at noon of the day. P0 was defined as the day the pups were born. Brains were dissected from E16.5, P0, P10, and P30 age mice (1 mouse per time point) and rinsed quickly in ice-cold phosphate-buffered saline (PBS). Brain tissues were embedded in optimal cutting temperature compound (OCT, Tissue Plus, Fisher) with embedding mold and frozen on dry ice. It should be noted that in an attempt to delineate embryonic CA regions and their layers, we removed the embryonic brain from the skull and embedding, resulting in slight flattening and altering of the original posture of this rather soft tissue. Embedded brain samples were stored at −80°C until further use. To sample the whole hippocampus across the spatial axes in sagittal and horizontal planes, 4 lateral-medial (sagittal) and dorsal-ventral (horizontal)

cryosections with an interval of 0.2 mm (for E16.5 brain), 0.3 mm (P0), 0.5 mm (P10), and 1.0 mm (P30) were collected. Brain sections were cut at 12 μm thickness and collected on super-frost slides (VWR) using a cryostat (Leica Microsystems, CM3050-S) and stored at −80°C until further use.

## Single-molecule RNA fluorescent in situ hybridization (smRNA-FISH)

Frozen slides containing unfixed brain sections were allowed to thaw at room temperature for 5 min before being fixed using ice-cold 4% paraformaldehyde (PFA). smRNA-FISH was performed on fixed slides using the multiplex RNAscope platform (Advanced Cell Diagnostics, 323100), according to the manufacturer instructions for fresh-frozen sections. The probes used for ISH were Mm-Tenm1-C3 (500641-C3), Mm-Tenm2-C2 (552671-C2), Mm-Tenm3 (411951), Mm-Tenm4-C4 (555491-C4), Mm-Lphn1 (#319331), Mm-Lphn2-C2 (319341-C2), Mm-Lphn3-C3 (317481-C3), Mm-Flrt1-C3 (555481-C3), Mm-Flrt2 (490291), and Mm-Flrt3-C2 (490301-C2). All probes and reagents were obtained from Advanced Cell Diagnostics. Hybridized slides were mounted using Prolong Gold anti-fade mounting medium containing DAPI for nuclear staining (Thermo Fisher, Cat# P36930).

## Automated image acquisition

RNA-ISH slides were imaged using the VS120 and VS200 automated slide scanner (Olympus) at 20× magnification. DAPI, FITC, TRITC, and Cy5 channels were used to acquire images. Exposure time for each channel was set manually and uniformly for all samples.

## Image analysis

The anatomical registration and quantification in this study was performed on sagittal sections covering mediolateral regions, mainly due to the availability of high-resolution reference annotations from the Allen Brain Atlas and Prenatal Mouse Brain reference atlas. There is no high-resolution annotated brain anatomy available for horizontal sections for various developmental stages. Therefore, we did not quantify images in the dorsal-ventral axis. Nevertheless, all the horizontal images (in addition to all sagittal images) are provided in the supplementary figures. Images of each sagittal brain section were individually exported to the JPG format, and the P10 and P30 images were registered using the Allen Mouse Brain Common Coordinate Framework (Allen Mouse Brain reference atlas). For the E16.5 and P0 images, *The Prenatal Mouse Brain Atlas* manual (2008 edition) [59] was used as a reference. Common structures between the ISH images and the reference atlases were individually identified, allowing brain regions to be delineated by hand using a tablet device and a stylus. Images were systematically registered one at a time to ensure a high degree of accuracy. Then, the registered images were transferred to a Java-based program, BREIN (Brain Region Expression Intensity Neuroanalysis) App [60], which runs a threshold algorithm to quantify each intensity value. The program outputs region-specific levels of gene expression by calculating the localized level of saturation of each color of ISH signal corresponding to each anatomical region. The resulting intensity values were normalized relative to the DAPI intensity values to account for differences in cell density across regions and time points. BREIN is available for download as a standalone software: https://med.stanford.edu/sudhoflab/science-resources/tools.html. The finalized data tables were plotted in MATLAB to generate bubble plots and GraphPad Prism (version 9.5.1) to generate the heat maps. For entorhinal cortex layer-specific expression quantification of Lphns, RNA-FISH intensities were measured at 5 different ROIs (region of interests) using Fiji (Image J) and average of values were calculated and plotted using GraphPad Prism.

## Datasets

To validate the smRNA-FISH data through RNA-seq data and to gain further insights into the expression patterns of Tenms, Lphns, and Flrts in hippocampal subclasses and non-neuronal cell populations, we analyzed publicly available RNA-seq datasets. Hipposeq (https://hipposeq.janelia.org/) datasets were used to find dorsoventral gene expression patterns in hippocampal principal neurons [33]. Allen Brain Map single-cell RNA-Seq data (https://celltypes.brain-map.org/rnaseq/mouse_ctx-hpf_10x) was used to identify overall expression patterns in different cell populations in the hippocampus and cortex [35]. The ASCOT database (Alternative Splicing & Gene Expression Summaries of Public RNA-Seq Data) (http://ascot.cs.jhu.edu/) was used to identify expression patterns in hippocampal projection classes. The Dropviz database (http://dropviz.org) was used to identify hippocampal subclass-specific expression [61]. The single-nucleus RNA-seq (sNuc-Seq) database (https://singlecell.broadinstitute.org/single_cell/study/SCP1/-single-nucleus-rna-seq-of-cell-diversity-in-the-adult-mouse-hippocampus-snuc-seq) was utilized to identify expression patterns in hippocampal neuronal and non-neuronal class and subclass (especially glial) cells. [37]. The Splicecode database (https://scheiffele-splice.scicore.unibas.ch/) was used to infer expression in inhibitory and excitatory populations from RiboTag translatomics datasets [38].

## Supporting information

**S1 Fig.** (A) Schematic diagram shows adult mouse hippocampal neural circuits with cells of CA1, CA3, DG, lateral, and medial entorhinal (LEC and MEC) areas. (B) Schematic diagram of hippocampus anatomy visualized through sagittal view for a spatial orientation. Transverse section of hippocampal region is shown to highlight the layers within the CA1 and subiculum regions. so–stratum oriens; sp–stratum pyramidale; sr–stratum radiatum; slm–stratum lacunosum-moleculare; mo–molecular layer; oml–outer molecular layer; mml–middle molecular layer; iml–inner molecular layer; sg–stratum granulare; po–polymorph layer; Sub–subiculum; PaS–parasubiculum; PrS–presubiculum; L1-L6 –layer 1–6; CA1, CA3 –cornu Ammonis 1, 3; DG–dentate gyrus; LEC and MEC–lateral and medial entorhinal cortex.
(TIF)

**S2 Fig. Microscopic images of sagittal sections from E16.5 mouse brains labeled with RNA in situ hybridization probes to detect Tenm1, Tenm2, Tenm3, and Tenm4 expression.** Location of each sagittal section in lateral-medial axis is shown through the cartoons on the left side. Images in this figure are re-used in Fig 2A.
(TIF)

**S3 Fig. Microscopic images of horizontal sections from E16.5 mouse brains labeled with RNA in situ hybridization probes to detect Tenm1, Tenm2, Tenm3, and Tenm4 expression.**
(TIF)

**S4 Fig. Microscopic images of sagittal sections of hippocampal region from P0 mouse brain labeled with RNA in situ hybridization probes to detect Tenm1, Tenm2, Tenm3, and Tenm4 expression.** Location of each sagittal section in lateral-medial axis is shown through the cartoons on the left side. Images in this figure are re-used in Fig 2B. S–subiculum; PrS–presubiculum; PaS–parasubiculum; CA1, CA3 –cornu Ammonis 1, 3; DG–dentate gyrus; MEnt–medial entorhinal cortex.
(TIF)

**S5 Fig. Microscopic images of horizontal sections of hippocampal region from P0 mouse brain labeled with RNA in situ hybridization probes to detect Tenm1, Tenm2, Tenm3, and Tenm4 expression.** S–subiculum; PrS–presubiculum; PaS–parasubiculum; CA1, CA3 –cornu Ammonis 1, 3; DG–dentate gyrus; MEnt–medial entorhinal cortex; Ect–ectorhinal area; PRh–perirhinal area.
(TIF)

**S6 Fig. Microscopic images of sagittal sections of hippocampal region from P10 mouse brain labeled with RNA in situ hybridization probes to detect Tenm1, Tenm2, Tenm3, and Tenm4 expression.** Location of each hippocampal sagittal section in lateral-medial axis is shown through the cartoons on the left side. Images in this figure are re-used in Fig 2E. S–subiculum; CA1, CA2, CA3 –cornu Ammonis 1, 2, 3; DG–dentate gyrus.
(TIF)

**S7 Fig. Microscopic images of horizontal sections of hippocampal region from P10 mouse brain labeled with RNA in situ hybridization probes to detect Tenm1, Tenm2, Tenm3, and Tenm4 expression.** Location of each hippocampal horizontal section in dorsoventral axis is shown through the cartoons on the left side. Images in this figure are re-used in Fig 7E. S–subiculum; PrS–presubiculum; PaS–parasubiculum; CA1, CA3 –cornu Ammonis 1, 3; DG–dentate gyrus; MEnt–medial entorhinal cortex; Ect–ectorhinal area; PRh–perirhinal area.
(TIF)

**S8 Fig. Microscopic images of sagittal sections of hippocampal region from P30 mouse brain labeled with RNA in situ hybridization probes to detect Tenm1, Tenm2, Tenm3, and Tenm4 expression.** Location of each hippocampal sagittal section in lateral-medial axis is shown through the cartoons on the left side. Images in this figure are re-used in Fig 2F. S–subiculum; PrS,–presubiculum; PaS–parasubiculum; Post–postsubiculum; CA1, CA2, CA3 –cornu Ammonis 1, 3; DG–dentate gyrus; MEnt–medial entorhinal cortex; Ect–ectorhinal area; PRh–perirhinal area.
(TIF)

**S9 Fig. Microscopic images of horizontal sections of hippocampal region from P30 mouse brain labeled with RNA in situ hybridization probes to detect Tenm1, Tenm2, Tenm3, and Tenm4 expression.** Location of each hippocampal horizontal section in dorsoventral axis is shown through the cartoons on the left side. Images in this figure are re-used in Fig 7E. Sub–subiculum; PrS,–presubiculum; PaS–parasubiculum; CA1, CA3 –cornu Ammonis 1, 3; DG–dentate gyrus; MEnt–medial entorhinal cortex; Ect–ectorhinal area; PRh–perirhinal area.
(TIF)

**S10 Fig.** Microscopic images of sagittal sections of hippocampal region from E16.5 (top panel) and P0 (bottom panel) mouse brains labeled with RNA in situ hybridization probes to detect Lphn1, Lphn2, and Lphn3 expression. Location of each sagittal section in lateral-medial axis is shown through the cartoons on the left side. Images in this figure are re-used in Figs 4A, 4B and 5D.
(TIF)

**S11 Fig.** Microscopic images of horizontal sections of hippocampal region from E16.5 (top panel) and P0 (bottom panel) mouse brains labeled with RNA in situ hybridization probes to detect Lphn1, Lphn2, and Lphn3 expression.
(TIF)

**S12 Fig.** Microscopic images of sagittal sections of hippocampal region from P10 (top panel) and P30 (bottom panel) mouse brains labeled with RNA in situ hybridization probes to detect Lphn1, Lphn2, and Lphn3 expression. Location of each hippocampal sagittal section in lateral-medial axis is shown through the cartoons on the left side. Images in this figure are re-used in Figs 4E, 4F and 5E. S–subiculum; PrS–presubiculum; PaS–parasubiculum; post–postsubiculum; CA1, CA3 –cornu Ammonis 1, 3; DG–dentate gyrus; MEnt–medial entorhinal cortex; Ect–ectorhinal area; PRh–perirhinal area.
(TIF)

**S13 Fig.** Microscopic images of horizontal sections of hippocampal region from P10 (top panel) and P30 (bottom panel) mouse brains labeled with RNA in situ hybridization probes to detect Lphn1, Lphn2, and Lphn3 expression. Location of each hippocampal horizontal section in dorsoventral axis is shown through the cartoons on the left side. Images in this figure are re-used in Figs 5E, 6A and 6B.
(TIF)

**S14 Fig. Heat map shows Tenm, Lphn, Flrt expression levels in various subclasses of GABAergic and glutamatergic cell types.** GABAergic cell marker Gad1 and Glutamatergic cell maker Slc17a7 (Vglut) expression levels are also shown in the bottom for quality control. Whole cortex and hippocampus scRNA-seq data obtained from Allen Brain Map - https://celltypes.brain-map.org/rnaseq/mouse_ctx-hpf_10x. Various cell types of GABAergic and glutamatergic neurons are indicated in different colors.
(TIF)

**S15 Fig.** (A) Bar graphs show expression levels of excitatory (Camk2a and Grik) and inhibitory (Sst, Scnn1a, Pvalb, and Vip) cell markers (as quality control) in respective cell populations as revealed by sequencing of ribosome-engaged total mRNA (RiboTag seq). Bar graphs show expression levels of Tenms (B), Lphns (C), and Flrts (D) in RiboTag inhibitory and excitatory cell populations in the hippocampus (HC) and cortex (CTX). RNA-seq data obtained from Splicecode database (https://scheiffele-splice.scicore.unibas.ch/). The data underlying graphs in A–D can be found in S1 Data.
(TIF)

**S16 Fig. Heat map shows expression of Tenms, Lphns, and Flrts in whole brain and non-neural tissues.** Note the high expression of Lphn1 in several non-neural tissues. RNA-seq data obtained from ASCOT database (http://ascot.cs.jhu.edu/). The data underlying this figure can be found in S1 Data. NAUC–Normalized Area Under the Curve.
(TIF)

**S1 Data. Excel file containing source data for results presented in Figs 3, 5, 6, 8, S9, S15 and S16.**
(XLSM)

## Acknowledgments

We thank Drs Justin Trotter, Matthew Pomrenze, Lu Chen, and Takao Hensch for helpful discussions and Margarita Artyukhova for critical reading of the manuscript.

## Author Contributions

**Conceptualization:** Kif Liakath-Ali, Thomas C. Südhof.

**Data curation:** Kif Liakath-Ali, Rebecca Refaee.

**Formal analysis:** Kif Liakath-Ali, Rebecca Refaee, Thomas C. Südhof.

**Funding acquisition:** Kif Liakath-Ali, Thomas C. Südhof.

**Investigation:** Kif Liakath-Ali, Thomas C. Südhof.

**Methodology:** Kif Liakath-Ali, Rebecca Refaee, Thomas C. Südhof.

**Project administration:** Kif Liakath-Ali, Thomas C. Südhof.

**Resources:** Kif Liakath-Ali, Thomas C. Südhof.

**Software:** Rebecca Refaee.

**Supervision:** Kif Liakath-Ali, Thomas C. Südhof.

**Validation:** Kif Liakath-Ali.

**Visualization:** Kif Liakath-Ali, Rebecca Refaee.

**Writing – original draft:** Kif Liakath-Ali, Thomas C. Südhof.

**Writing – review & editing:** Kif Liakath-Ali, Rebecca Refaee, Thomas C. Südhof.

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
