## [Editor Report · Decision Letter 0]

6 Jun 2023

Dear Dr Liakathali, 

Thank you for submitting your manuscript entitled "Cartography of Teneurin and Latrophilin Expression Reveal Spatiotemporal Axis Heterogeneity in the Hippocampus" for consideration as a Methods and Resources by PLOS Biology.

Your manuscript has now been evaluated by the PLOS Biology editorial staff as well as by an academic editor with relevant expertise and I am writing to let you know that we would like to send your submission out for external peer review.

Once your full submission is complete, your paper will undergo a series of checks in preparation for peer review. After your manuscript has passed the checks it will be sent out for review. To provide the metadata for your submission, please Login to Editorial Manager (https://www.editorialmanager.com/pbiology) within two working days, i.e. by Jun 08 2023 11:59PM.

Kind regards,

Christian

Christian Schnell, PhD

Senior Editor

PLOS Biology

cschnell@plos.org

---

## [Decision Letter · Decision Letter 1]

13 Jul 2023

Dear Dr Liakath-Ali,

Thank you for your patience while your manuscript "Cartography of Teneurin and Latrophilin Expression Reveal Spatiotemporal Axis Heterogeneity in the Hippocampus" was peer-reviewed at PLOS Biology. Your manuscript has been evaluated by the PLOS Biology editors, an Academic Editor with relevant expertise, and by several independent reviewers.

As you will see in the reviewer reports, which can be found at the end of this email, although the reviewers find the work potentially interesting, they have also raised a substantial number of important concerns. Based on their specific comments and following discussion with the Academic Editor, it is clear that a substantial amount of work would be required to meet the criteria for publication in PLOS Biology. However, given our and the reviewer interest in your study, we would be open to inviting a comprehensive revision of the study that thoroughly addresses all the reviewers' comments. Given the extent of revision that would be needed, we cannot make a decision about publication until we have seen the revised manuscript and your response to the reviewers' comments. Your revised manuscript would need to be seen by the reviewers again, but please note that we would not engage them unless their main concerns have been addressed. 

Having discussed the reviews with the Academic Editor, we think you should in particular address Reviewer 3's concerns and collect new data to strengthen the resource value of your study. 

We appreciate that these requests represent a great deal of extra work, and we are willing to relax our standard revision time to allow you 6 months to revise your study. Please email us (plosbiology@plos.org) if you have any questions or concerns, or envision needing a (short) extension.

**IMPORTANT - SUBMITTING YOUR REVISION**

*Resubmission Checklist*

*Published Peer Review*

*PLOS Data Policy*

*Blot and Gel Data Policy*

Kind regards,

Christian

Christian Schnell, PhD

Senior Editor

PLOS Biology

cschnell@plos.org

REVIEWS:

Reviewer #1: Liakath-Ali et al present a well-written paper with beautiful figures that presents the spatial and temporal distribution of synaptic adhesion molecules, two of which have been shown to be involved in embryonic hippocampus development. The validation of their in situ hybridization using RNAseq databases is clever and provides a nice example of the utility of these databases in studies like this. That being said, this article is not clearly presented as a methods paper, and, as such, my comments are about framing their results in the broader picture of hippocampus development because that is how they have begun to frame the work. I think that if the intention is for this paper to be a methods paper, the authors should reframe their work in that regard.

I have a few comments that I think would increase the strength of the paper. 

1) In the abstract the authors present clear logic as to why the focus of the paper is on tenuerins and latrophilins (because of the role of Tenm3 and Lphn2 in embryonic hippocampus development), and this should be carried into the Introduction. Adding this information to the introduction would make the reasoning behind the study more robust and cohesive.

The authors also note that these SAMs are evolutionarily conserved during development? It would be interesting to comment if these are the only SAMs that are evolutionarily conserved, and also to note that connectivity of the hippocampal-parahippocampal regions is itself heavily conserved across species. This is likely a point for the discussion, but it might help highlight the relevance and role of these SAMs in the hippocampal-parahippocampal development. 

2) The description of the development of the hippocampus in the introduction could be clarified and should include references. The description that neurogenesis primarily occurs in the dentate gyrus and subventricular zone is something that is described a lot for adult neurogenesis, and is even described for adult neurogenesis in the reference provided by the author (19). However, because the dentate gyrus is a later emerging structure, it is unclear why the authors describe this in the adult terms. For example, many developmental biologists would instead say that origin of hippocampus neurons (neurogenesis) in the developing embryo includes glutamatergic cells produced locally by progenitors in the ventricular zone of the primordial hippocampal area (Cossart and Khazipov, Physiological Reviews 2022). Interneurons arise from the median ganglionic eminence. Additonally, I understand that the current study is performed in mice, but in the rat, about 90 percent of the dentate granule cells are born after birth of the pup, therefore, the current language used to describe neurogenesis in the hippocampus seems misleading, and I personally think it would be better to phrase the origin of hippocampus neurons in the more common developmental terms.

3) The authors introduce very broadly the role of the dorsal and ventral hippocampal poles with very few references on the topic. For example, the few references that the authors include are not the primary references for the connectivity of the hippocampus with the structures listed in the introduction. It would be good to read and include some primary references.

Methods

1) The authors do not indicate how the timed breedings were performed to collect E16.5 embryos, or how postnatal day 0 was determined. If timed breeders were ordered from Charles River, the authors should confirm that they have the gestational days they expected. My experience with Charles River is that they call the day the plug is observed day 1 (instead of the widely accepted 0.5) and therefore, timed breeders purchased from Charles River are actually a day earlier in the gestational period. 

2) It is confusing and unclear what the authors are referring to when they say that "four lateral-medial (sagittal) and dorsal-ventral (horizontal) cryosections with an interval of 0.2 mM (for E16.5), 0.5 mM (P0), 0.75 mM (P10) and 1.0 mM (P30) were collected. mM what? Is this in reference to the concentration of the solution that the slices were incubated in? Is there a typo in the units such that it should be mm? 

Results

Figure 2

The authors should indicate on the embryonic tissue what they are delineating as each hippocampal region. It would be useful for the authors to include a schematic of the developing brain at E16.5 with the CA1, CA2, and CA3 regions identified. 

It also seems that from the DAPI stain, the authors should be able to delineate the layers of CA1-CA3 in the P0 brain, though this is hard to see with the SAMs overlaid as well.

It is also unfortunate that the color scheme representing the layers of the hippocampus has a high level of overlap with the colors used to label the tenuerins in the ISH slides. The authors should consider changing the color scheme of the hippocampus layers.

Figure 3

I think the authors did a nice job of representing the data in panels A through D. I like the representation of the mean intensity and how they go about translating the regions from age to age. This is easy to follow.

Challenging to follow is the lateral to medial staining (until I got to Figure 5, where the lateral to medial sections are easier to identify and follow). In Figure 3, the medial and lateral sections do not appear very different, whereas the authors present the full range of lateral to medial sections used in Figure 5. A schematic of the laterality represented in each pane for panel E would be helpful (schematic of lateral to medial sections). 

Figure 6

It is unclear to me why the figure title refers to the expression is as a gradient. The staining in the respective layers looks consistent along the dorsal-ventral axis. 

When describing the data in the results for this figure, the authors state that "With horizontal sections, we were able to capture two dorsoventral planes of EC." It is really unclear what they mean by this. What are the two dorsoventral planes?

That authors go on to say that "In P30, the ventral-most EC (caudomedial entorhinal cortex) regions showed striking enrichment of Lphns in multiple layers with high levels of Lphn1 in L2 and almost no expression of all Lphns in L1." Thinking forward about the discussion, the authors should indicate where in the cell these proteins are found. Layer 1 is a relatively cell-free layer that consists mostly of fibers, so it isn't surprising to not see label here when all the other RNA label appears to be seen in cell bodies. 

Discussion

1) On page 12, lines 6-8, the authors should provide a reference for the statement that the selective expression of Tenm, Lphn and Flrt paralogs outside of the CNS suggests that they may play similar cell-adhesion roles in relevant tissues. 

2) On page 13, lines 5-7, the authors should provide a reference for the critical period for experience-dependent plasticity in CA1, and they should additionally state what this time period is. The authors find that Tenm1 peaks at P10 and likely the authors mean that this coincides with the critical period for experience-dependent plasticity, but they also should recognize that the window of P10 to P30 is quite a lot of time when it comes to development and stating the critical window would help put their finding into perspective. 

3) The terminology using "mediolateral" is confusing, especially because the authors try to translate it to dorsal ventral using the HippoSeq data (Figure 7). Why not simply delineate the dorsal and ventral portions of the hippocampus from the sagittal sections to begin with? This would also help with the nomenclature that is used by others with respect to medial and lateral being based on looking at sections from the horizontal or coronal orientations. It seems that the authors should have the ability to delineate the dorsal and ventral portions of the hippocampus from the sagittal views based on the Supplementary Figures 1, 6 and 8. In taking the average intensity of the sections base on the laterality, there certainly would be an average of dorsal and ventral portions of the hippocampus in some sections that could hide increased intensity in either pole. If the point is to be able to compare across ages, the authors should consider doing both the dorsal-ventral delineations and the laterality, recognizing that in the case of the laterality, the embryonic hippocampus doesn't have the same amount of curvature and therefore the direct comparisons are difficult to make. Importantly, the authors could include more information in the methods to describe why they decided to take measurements and average the way they did.

4) Finally, the authors could do more to place their results into the larger framework of hippocampus development and function. As written, there is information about expression of genes in the dentate gyrus and CA1 that may relate to function in these regions, but the authors do not say much in regard to the role of the SAMs in these processes. Furthermore, the authors suggest that the reader could use the data to generate hypotheses, but it seems like the authors could also suggest some hypotheses, as they are experts in SAMs, in order to help place their results into perspective. 

Minor points:

1) There are a number of abbreviations that should be spelled out (e.g. FPKM in the y-axis of Figure 7, NAUC in Figure 8) for the reader with little RNAseq experience.

2) The use of deep and superficial in the Supplementary Figure 2 is also confusing. Deep and superficial typically are used for terminology of looking at the cells in the CA layers. Why not call it dorsal and ventral here?

3) Again, in the supplemental figures it would be nice to see a schematic of what was used to delineate the hippocampal regions in the embryonic hippocampus, in particular in the very lateral sections. 

Reviewer #2: The new manuscript provides explicit spatiotemporal information on how teneurins and latrophilins are expressed in the brain during development and after birth. Since these molecules (as cell adhesion molecules) are involved in formin

---

## [Decision Letter · Decision Letter 2]

23 Feb 2024

Dear Dr Liakath-Ali,

Thank you for your patience while we considered your revised manuscript "Cartography of Teneurin and Latrophilin Expression Reveal Spatiotemporal Axis Heterogeneity in the Hippocampus" for publication as a Methods and Resources at PLOS Biology. This revised version of your manuscript has been evaluated by the PLOS Biology editors, the Academic Editor and one of the original reviewers.

Based on the reviews and on our Academic Editor's assessment of your revision, we are likely to accept this manuscript for publication, provided you satisfactorily address the following data and other policy-related requests:

* We would like to suggest a different title to improve accuracy: "Cartography of Teneurin and Latrophilin Expression Reveals Spatiotemporal Axis Heterogeneity in the Mouse Hippocampus during Development"

* Please add the links to the funding agencies in the Financial Disclosure statement in the manuscript details.

* Please include the full name of the IACUC/ethics committee that reviewed and approved the animal care and use protocol/permit/project license. Please also include an approval number.

* Please include the specific national or international regulations/guidelines to which your animal care and use protocol adhered. Please note that institutional or accreditation organization guidelines (such as AAALAC) do not meet this requirement.

* Please mention in the data availability statement where the custom-made program can be found. If you decide to use github for code and/or data deposition, please assign a DOI so that the repository is citable and versioned for your paper. Zenodo is one of the available tools for this.

DATA POLICY:

Regardless of the method selected, please ensure that you provide the individual numerical values that underlie the summary data displayed in the following figure panels as they are essential for readers to assess your analysis and to reproduce it. Thank you for providing already most of the source data. Could please also add the data for Figure 8A, 8B, 8C and add the information where the data underlying this figure can be found to the corresponding figure legends? For example, "Source data can be found in S1 Data."

CODE POLICY

Per journal policy, as the code that you have generated is important to support the conclusions of your manuscript, we require that you make it available without restrictions upon publication. Please ensure that the code is sufficiently well documented and reusable, and that your Data Statement in the Editorial Manager submission system accurately describes where your code can be found.

We expect to receive your revised manuscript within two weeks. 

*Published Peer Review History*

*Press*

Sincerely,

Christian

Christian Schnell, PhD, 

Senior Editor

cschnell@plos.org

PLOS Biology

Reviewer remarks:

Reviewer #3: I have reviewed the manuscript and am pleased with the authors' response and the new data provided to address all the questions.

---

## [Editor Report · Decision Letter 3]

26 Mar 2024

Dear Dr Liakath-Ali,

Thank you for the submission of your revised Methods and Resources "Cartography of Teneurin and Latrophilin Expression Reveals Spatiotemporal Axis Heterogeneity in the Mouse Hippocampus during Development" for publication in PLOS Biology. On behalf of my colleagues and the Academic Editor, Eunjoon Kim, I am pleased to say that we can in principle accept your manuscript for publication, provided you address any remaining formatting and reporting issues. These will be detailed in an email you should receive within 2-3 business days from our colleagues in the journal operations team; no action is required from you until then. Please note that we will not be able to formally accept your manuscript and schedule it for publication until you have completed any requested changes.

As you address the formatting requests to come, please also attend to these final editorial requests:

* Please cite the location of the data in the figure legend of Figure 8. For example, 'The data underlying this Figure can be found in S1 Data'”. 

* Please add information in the corresponding supplementary figures when panels have been re-used in the main figures (you mention this already in the figure legends from the main figures but not the other way round). 

PRESS

Sincerely, 

Christian

Christian Schnell, PhD

Senior Editor

PLOS Biology

cschnell@plos.org